# Secoiridoid Glucosides and Anti-Inflammatory Constituents from the Stem Bark of *Fraxinus chinensis*

**DOI:** 10.3390/molecules25245911

**Published:** 2020-12-14

**Authors:** Hao-Chiun Chang, Shih-Wei Wang, Chin-Yen Chen, Tsong-Long Hwang, Ming-Jen Cheng, Ping-Jyun Sung, Kuang-Wen Liao, Jih-Jung Chen

**Affiliations:** 1Department of Orthopaedics, MacKay Memorial Hospital, Taipei 10449, Taiwan; Changhaochiun@gmail.com; 2Ph.D. Degree Program of Biomedical Science and Engineering, National Chiao Tung University, Hsinchu City 30068, Taiwan; liaonms@g2.nctu.edu.tw; 3Department of Medicine, MacKay Medical College, New Taipei City 25242, Taiwan; shihwei@mmc.edu.tw; 4Graduate Institute of Natural Products, Kaohsiung Medical University, Kaohsiung 80708, Taiwan; pjsung@nmmba.gov.tw; 5Graduate Institute of Pharmaceutical Technology, Tajen University, Pingtung 90741, Taiwan; jjc8506674@gmail.com; 6Graduate Institute of Natural Products, School of Traditional Chinese Medicine, College of Medicine, Chang Gung University, Taoyuan 33303, Taiwan; htl@mail.cgu.edu.tw; 7Research Center for Food and Cosmetic Safety, Graduate Institute of Health Industry Technology, College of Human Ecology, Chang Gung University of Science and Technology, Taoyuan 33303, Taiwan; 8Department of Anesthesiology, Chang Gung Memorial Hospital, Taoyuan 333, Taiwan; 9Bioresource Collection and Research Center (BCRC), Food Industry Research and Development Institute (FIRDI), Hsinchu 30062, Taiwan; cmj@firdi.org.tw; 10National Museum of Marine Biology and Aquarium, Pingtung 94450, Taiwan; 11Institute of Molecular Medicine and Bioengineering, National Chiao Tung University, Hsinchu City 30068, Taiwan; 12Faculty of Pharmacy, School of Pharmaceutical Sciences, National Yang-Ming University, Taipei 11221, Taiwan; 13Department of Medical Research, China Medical University Hospital, China Medical University, Taichung 40402, Taiwan

**Keywords:** *Fraxinus chinensis*, Oleaceae, stem bark, secoiridoid, anti-inflammatory activity

## Abstract

Qin Pi (*Fraxinus chinensis* Roxb.) is commercially used in healthcare products for the improvement of intestinal function and gouty arthritis in many countries. Three new secoiridoid glucosides, (8*E*)-4′′-*O*-methylligstroside (**1**), (8*E*)-4′′-*O*-methyldemethylligstroside (**2**), and 3′′,4′′-di-*O*-methyl-demethyloleuropein (**3**), have been isolated from the stem bark of *Fraxinus chinensis*, together with 23 known compounds (**4**–**26**). The structures of the new compounds were established by spectroscopic analyses (1D, 2D NMR, IR, UV, and HRESIMS). Among the isolated compounds, (8*E*)-4′′-*O*-methylligstroside (**1**), (8*E*)-4′′-*O*-methyldemethylligstroside (**2**), 3′′,4′′-di-*O*-methyldemethyloleuropein (**3**), oleuropein (**6**), aesculetin (**9**), isoscopoletin (**11**), aesculetin dimethyl ester (**12**), fraxetin (**14**), tyrosol (**21**), 4-hydroxyphenethyl acetate (**22**), and (+)-pinoresinol (**24**) exhibited inhibition (IC_50_ ≤ 7.65 μg/mL) of superoxide anion generation by human neutrophils in response to formyl-L-methionyl-L-leuckyl-L-phenylalanine/cytochalasin B (fMLP/CB). Compounds **1**, **9**, **11**, **14**, **21**, and **22** inhibited fMLP/CB-induced elastase release with IC_50_ ≤ 3.23 μg/mL. In addition, compounds **2**, **9**, **11**, **14**, and **21** showed potent inhibition with IC_50_ values ≤ 27.11 μM, against lipopolysaccharide (LPS)-induced nitric oxide (NO) generation. The well-known proinflammatory cytokines*,* tumor necrosis factor-alpha (TNF-α) and interleukin 6 (IL-6), were also inhibited by compounds **1**, **9**, and **14**. Compounds **1**, **9**, and **14** displayed an anti-inflammatory effect against NO, TNF-α, and IL-6 through the inhibition of activation of MAPKs and IκBα in LPS-activated macrophages. In addition, compounds **1**, **9**, and **14** stimulated anti-inflammatory M2 phenotype by elevating the expression of arginase 1 and Krüppel-like factor 4 (KLF4). The above results suggested that compounds **1**, **9**, and **14** could be considered as potential compounds for further development of NO production-targeted anti-inflammatory agents.

## 1. Introduction

*Fraxinus chinensis* Roxb. (Oleaceae) is a deciduous tree distributed in China, Japan, Korea, Russia, and Vietnam [1]. Its stem bark, called “Qin Pi”, is used as a health food or herbal supplement for improving intestinal function in Asia and America. The Oleaceae family is a rich source of secoiridoid glucosides [2]. A number of secoiridoid glucosides [2,3,4,5,6], coumarins [5,6], phenylpropenoids [5,6], lignans [6], and benzofuran derivatives [6] have been reported from the genus *Fruxinus*. These derivatives have been reported to exhibit several biological activities, such as antidiabetic [4], anti-inflammatory [4], immunosuppressive [4], anticancer [4], and quinone reductase-inducing activities [6]. *F. chinensis* has been found to be an active material by screening for anti-inflammatory effect of many natural sources. Three new secoiridoid glucosides, (8*E*)-4′′-*O*-methylligstroside (**1**), (8*E*)-4′′-*O*-methyldemethylligstroside (**2**), and 3′′,4′′-di-*O*-methyldemethyloleuropein (**3**), and 23 known compounds (**4**–**26**) have been isolated and confirmed from the stem bark of *F. chinensis*. This report depicts the structural elucidation of three new compounds **1**–**3** and the inhibitory activities of all isolated compounds against fMLP/CB-induced O_2_^•−^ and elastase release and against LPS-induced NO generation.

## 2. Results and Discussion

### 2.1. Isolation and Structural Elucidation

Separation of the EtOAc-soluble fraction of an MeOH extract of stem bark of *F. chinensis* by silica gel chromatography and preparative thin-layer chromatography (TLC) afforded three new (**1**–**3**) and 23 known compounds (**4**–**26**) (Figure 1).

Compound **1** was obtained as yellowish oil and the molecular formula was determined to be C_26_H_34_O_12_ by ESI-MS [*m*/*z* 561 [M + Na]^+^] (Appendix A) and HR-ESI-MS [*m*/*z* 561.1950 [M + Na]^+^ (calcd for C_26_H_34_NaO_12_, 561.1948)] (Appendix A). The IR spectrum showed the presence of hydroxyl (3402 cm^−1^) and carbonyl (1727 and 1709 cm^−1^) groups. Analysis of the ^1^H (Table 1 and Appendix A) and ^13^C NMR (Table 2 and Appendix A) data of **1** revealed signals for a 4-methoxyphenethoxy group [δ_H_ 2.85 (2H, t, *J* = 7.0 Hz, H-β), 3.76 (3H, s, OMe-4′′), 4.12, 4.24 (each 1H, each dt, *J* = 10.5, 7.0 Hz, H-α), 6.85 (2H, d, *J =* 9.0 Hz, H-3′′ and H-5′′), 7.15 (2H, d, *J =* 9.0 Hz, H-2′′ and H-6′′); δ_C_ 35.2 (C-β), 55.9 (OMe-4′′), 67.0 (C-α), 115.1 (C-3′′ and C-5′′), 131.2 (C-2′′ and C-6′′), 131.5 (C-1′′), 160.0 (C-4′′)], a secoiridoid moiety [δ_H_ 1.62 (3H, dd, *J* = 7.0, 1.0 Hz, H-10), 2.44 (1H, dd, *J* = 14.0, 9.5 Hz, H-6), 2.69 (1H, dd, *J* = 14.0, 5.0 Hz, H-6), 3.71 (3H, s, OMe-11), 3.95 (1H, dd, *J* = 9.5, 5.0 Hz, H-5), 5.92 (1H, br s, H-1), 6.06 (1H, br q, *J* = 7.0 Hz, H-8), 7.51 (1H, s, H-3); δ_C_ 13.7 (C-10), 32.0 (C-5), 41.3 (C-6), 52.1 (CH_3_OCO-4), 95.3 (C-1), 109.5 (C-4), 125.1 (C-8), 130.5 (C-9), 155.3 (C-3), 168.9 (CH_3_OCO-4), 173.4 (C-7)], and a β-glucose moiety [δ_H_ 3.28–3.36 (3H, m, H-2′, H-4′, and H-5′), 3.42 (1H, dd, *J* = 8.5, 8.5 Hz, H-3′), 3.66 (1H, dd, *J* = 12.0, 5.5 Hz, H-6′), 3.88 (1H, dd, *J* = 12.0, 1.5 Hz, H-6′), 4.80 (1H, d, *J* = 7.5 Hz, H-1′); δ_C_ 62.9 (C-6′), 71.7 (C-4′), 74.9 (C-2′), 78.1 (C-3′), 78.6 (C-5′), 101.0 (C-1′)]. These data were nearly identical with those of (8*E*)-ligstroside (**5**) [7], except that a methoxy group [δ_H_ 3.76 (3H, s); δ_C_ 55.9] at C-4′′ of **1** replaced the 4′′-hydroxy group of (8*E*)-ligstroside (**5**) [7]. This was supported by NOESY correlations between OMe-4′′ (δ_H_ 3.76) and H-3′′/H-5′′ (δ_H_ 6.85) and by HMBC correlation between OMe-4′′ (δ_H_ 3.76) and C-4′′ (δ_C_ 160.0) (Figure 2). The *E*-configuration at C-8 was comfirmed by NOESY correlation between H-5 and H-10. In the NOESY spectrum, H-1 (δ 5.92) had the correlation with H-6 (δ 2.44) and had no correlation with H-5 (δ 3.95), which indicated the relative configurations of H-1 and H-5 as α and β, respectively. The position of each substituent was supported by NOESY correlations (Figure 2) between H-1 (δ_H_ 5.92)/H-6 (δ_H_ 2.44), H-1 (δ_H_ 5.92)/H-8 (δ_H_ 6.06), H-5 (δ_H_ 3.95)/H-10 (δ_H_ 1.62), H-1′ (δ_H_ 4.80)/H-3′ (δ_H_ 3.42), H-β (δ_H_ 2.85)/H-2′′ (δ_H_ 7.15), and H-3′′ (δ_H_ 6.85)/OMe-4′′ (δ_H_ 3.76) and by HMBC correlation (Figure 2) between H-1 (δ_H_ 5.92)/C-8 (δ_C_ 125.1), H-1 (δ_H_ 5.92)/C-1′ (δ_C_ 101.0), H-3 (δ_H_ 7.51)/C-1 (δ_C_ 95.3), H-3 (δ_H_ 7.51)/C-5 (δ_C_ 32.0), H-5 (δ_H_ 3.95)/C-7 (δ_C_ 173.4), H-5 (δ_H_ 3.95)/C-11 (δ_C_ 168.9), H-10 (δ_H_ 1.62)/C-9 (δ_C_ 130.5), OMe-11 (δ_H_ 3.71)/C-11 (δ_C_ 168.9), H-α (δ_H_ 4.12)/C-7 (δ_C_ 173.4), H-α (δ_H_ 4.12)/C-1′′ (δ_C_ 131.5), H-β (δ_H_ 2.85)/C-2′′,6′′ (δ_C_ 131.2), H-3′′,5′′ (δ_H_ 6.85)/C-1′′ (δ_C_ 131.5), and OMe-4′′ (δ_H_ 3.76)/C-4′′ (δ_C_ 160.0). The full assignment of ^1^H and ^13^C NMR resonances was supported by DEPT (Appendix A), ^1^H–^1^H COSY (Appendix A), NOESY (Appendix A), HMBC (Appendix A), and HSQC (Appendix A) spectral analyses. According to the above data, the structure of **1** was elucidated as (8*E*)-4′′-*O*-methylligstroside.

Compound **2** was obtained as amorphous powder. The ESI-MS (Appendix A) afforded a sodium adduct ion [M + Na]^+^ at *m/z* 547, implying a molecular formula of C_25_H_32_O_12_, which was confirmed by the HR-ESI-MS mass spectrum (*m/z* 547.1787 [M + Na]^+^, calcd for C_25_H_32_O_12_Na, 547.1791) (Appendix A). The presence of hydroxyl (3334 cm^−1^) and carbonyl (1728 and 1707 cm^−1^) groups were evident from the IR spectrum. The ^1^H (Table 1 and Appendix A) and ^13^C NMR (Table 2 and Appendix A) data of **2** were very similar to those of demethylligstroside [3], except that a methoxy group [δ_H_ 3.76 (3H, s)] at C-4′′ in **2** replaced the 4′′-hydroxy group of demethylligstroside [3]. This was supported by NOESY correlations between OMe-4′′ (δ_H_ 3.76) and H-3′′/H-5′′ (δ_H_ 6.85) and by HMBC correlation between OMe-4′′ (δ_H_ 3.76) and C-4′′ (δ_C_ 160.0) (Figure 3). The relative configuration of **2** was assigned by NOESY spectrum, which showed correlation between H-1 (δ 5.87) and H-6 (δ 2.41), suggesting that H-5 was on the β configuration, and H-1 was on the α configuration. The *E*-configuration at C-8 was comfirmed by NOESY correlation between H-5 and H-10. The full assignment of ^1^H and ^13^C NMR resonances was supported by DEPT (Appendix A), ^1^H–^1^H COSY (Appendix A), NOESY (Figure 3 and Appendix A), HMBC (Figure 3 and Appendix A), and HSQC (Appendix A) spectral analyses. Thus, the structure of **2** was established as shown in Figure 1, and named (8*E*)-4′′-*O*-methyldemethylligstroside. 

Compound **3** was isolated as amorphous powder. Its molecular formula, C_26_H_34_O_13_, was determined on the basis of the positive ESI-MS at *m/z* 577 [M + Na]^+^ (Appendix A) and HR-ESI-MS at *m/z* 577.1892 [M + Na]^+^ (calcd 577.1897) (Appendix A) and was supported by the ^1^H, ^13^C, and DEPT NMR data. The IR absorption bands of **3** revealed the presence of hydroxyl (3350 cm^−1^) and carbonyl (1721 and 1698 cm^−1^) functions. The ^1^H (Table 1 and Appendix A) and ^13^C NMR (Table 2 and Appendix A) data of **3** were similar to those of **2**, except that a 3,4-dimethoxyphenyl group [δ_H_ 3.80 (3H, s, OMe-4′′), 3.82 (3H, s, OMe-3′′), 6.79 (1H, dd, *J* = 8.5, 2.0 Hz, H-6′′), 6.86 (1H, d, *J* = 2.0 Hz, H-2′′), 6.88 (1H, d, *J* = 8.5 Hz, H-5′′); δ_C_ 56.7 (OMe-3′′), 56.7 (OMe-4′′), 113.3 (C-5′′), 114.1 (C-2′′), 122.6 (C-6′′), 132.5 (C-1′′), 149.3 (C-4′′), 150.5 (C-3′′)] at C-β in **3** replaced the 4-methoxyphenyl group at C-β of **2**. This was supported by NOESY correlations between OMe-3′′ (δ_H_ 3.82) and H-2′′ (δ_H_ 6.86) and by HMBC correlation between OMe-3′′ (δ_H_ 3.82) and C-3′′ (δ_C_ 150.5) (Figure 4). The relative configuration of **3** was assumed to be the same as that of **2** based on the NOESY correlation between H-1 (δ_H_ 5.86) and H-6 (δ_H_ 2.41). The *E*-configuration at C-8 was comfirmed by NOESY correlation between H-5 and H-10. The full assignment of ^1^H and ^13^C NMR resonances was further confirmed by DEPT (Appendix A), ^1^H–^1^H COSY (Appendix A), NOESY (Figure 4 and Appendix A), HMBC (Figure 4 and Appendix A), and HSQC (Appendix A) data. Consequently, the structure of compound **3** was established as 3′′,4′′-di-*O*-methyldemethyloleuropein.

### 2.2. Structure Identification of the Known Isolates

The known isolates were readily identified by a comparison of their physical and spectroscopic data (UV, IR, ^1^H NMR, [α]_D_, and MS) with those of authentic samples or literature values. They include a pyran derivative, fraxilatone (**4**) [8], four secoiridoids, (8*E*)-ligstroside (**5**) [7,9], oleuropein (**6**) [9], (8*E*)-3′′,4′′-di-*O*-methyloleuropein (**7**) [10], and oleoside methyl ester (**8**) [11], seven coumarins, aesculetin (**9**) [12], scopoletin (**10**) [12,13], isoscopoletin (**11**) [12,14], aesculetin dimethyl ester (**12**) [12,15], fraxidin (**13**) [16,17], fraxetin (**14**) [18], and umbelliferone (**15**) [19], five phenylpropanoids, methyl isoferulate (**16**) [20], methyl ferulate (**17**) [21], methyl 3,4-dimethoxycinnamate (**18**) [22], methyl (*E*)-*p*-coumarate (**19**) [23], and (*E*)-ferulaldehyde (**20**) [24], two phenylethanoids, tyrosol (**21**) [25] and 4-hydroxyphenethyl acetate (**22**) [26], a benzenoid, *p*-hydroxybenzaldehyde (**23**) [27], two lignans, (+)-pinoresinol (**24**) [28] and (+)-salicifoliol (**25**) [29], and a α-tocopheranoid: α-tocopheryl quinone (**26**) [30].

### 2.3. Biological Studies

Reactive oxygen species (ROS) (e.g., hydrogen peroxide and superoxide anion (O_2_^•−^)) and granule proteases (e.g., elastase, proteinase-3, and cathepsin G) produced by human neutrophils are involved in the pathogenesis of a variety of inflammatory diseases [31,32,33]. The effects on neutrophil proinflammatory responses of isolated compounds from the stem bark of *F. chinensis* were evaluated by suppressing fMLP/CB-induced superoxide radical anion (O_2_^•−^) generation and elastase release by human neutrophils. The inhibitory activity data on human neutrophil proinflammatory responses are shown in Table 3. Diphenyleneiodonium and phenylmethylsulfonyl fluoride were used as positive controls for O_2_^•−^ generation and elastase release, respectively. From the results of our biological tests, the following conclusions can be drawn: (a) (8*E*)-4′′-*O*-methylligstroside (**1**), (8*E*)-4′′-*O*-methyldemethylligstroside (**2**), 3′′,4′′-di-*O*-methyldemethyloleuropein (**3**), oleuropein (**6**), aesculetin (**9**), isoscopoletin (**11**), aesculetin dimethyl ester (**12**), fraxetin (**14**), tyrosol (**21**), 4-hydroxyphenethyl acetate (**22**), and (+)-pinoresinol (**24**) showed potent inhibition (IC_50_ ≤ 7.65 μg/mL) of O_2_^•−^ generation by neutrophils in response to fMLP/CB; (b) (8*E*)-4′′-*O*-methylligstroside (**1**), aesculetin (**9**), isoscopoletin (**11**), fraxetin (**14**), tyrosol (**21**), and 4-hydroxyphenethyl acetate (**22**) displayed potent inhibition (IC_50_ ≤ 3.23 μg/mL) against fMLP-induced elastase release; (c) secoiridoid glucoside, (8*E*)-4′′-*O*-methylligstroside (**1**) (with 4′′-methoxy group) displayed more effective inhibition than its analogue, (8*E*)-ligstroside (**5**) (with 4′′-hydroxy group) against fMLP-induced O_2_^•−^ generation and elastase release; (d) among the 6,7-disubstituted coumarin derivatives, aesculetin (**9**) (with 6,7-dihydroxy groups) exhibited more effective inhibition than its analogues, scopoletin (**10**) (with 7-hydroxy-6-methoxy groups), isoscopoletin (**11**) (with 6-hydroxy-7-methoxy groups), and aesculetin dimethyl ester (**12**) (with 6,7-dimethoxy groups) against fMLP-induced O_2_^•−^ generation; (e) among the 6,7,8-trisubstituted coumarin derivatives, fraxetin (**14**) (with 7,8-dihydroxy-6-methoxy groups) displayed more effective inhibition than its analogue, fraxidin (**13**) (with 8-hydroxy-6,7-dimethoxy groups) against fMLP-induced O_2_^•−^ generation and elastase release; (f) (8*E*)-4′′-*O*-methylligstroside (**1**) and fraxetin (**14**) were the most effective among the isolated compounds, with IC_50_ values of 0.08 ± 0.01 and 0.50 ± 0.10 μg/mL, respectively, against fMLP-induced O_2_^•−^ generation and elastase release.

Nitric oxide (NO) is a mediator in the inflammatory response involved in host defense. The anti-inflammatory effects of the compounds isolated from the stem bark of *F. chinensis* were also evaluated by suppressing lipopolysaccharide (LPS)-induced NO generation in macrophage cell line RAW264.7. The inhibitory activity data of the isolates **1**–**26** on NO generation by macrophages are shown in Table 4. Quercetin was used as the positive control. From the results of our anti-inflammatory assays, the following conclusions can be drawn: (a) (8*E*)-4′′-*O*-methyldemethylligstroside (**2**), aesculetin (**9**), isoscopoletin (**11**), fraxetin (**14**), and tyrosol (**21**) showed potent inhibition with IC_50_ values ≤ 27.11 μM, against lipopolysaccharide (LPS)-induced nitric oxide (NO) generation; (b) secoiridoid glucoside, (8*E*)-4′′-*O*-methylligstroside (**1**) (with 4′′-methoxy group) displayed more effective inhibition than its analogue, (8*E*)-ligstroside (**5**) (with 4′′-hydroxy group) against LPS-induced NO generation; (c) among the 6,7-disubstituted coumarin derivatives, aesculetin (**9**) (with 6,7-dihydroxy groups) and isoscopoletin (**11**) (with 6-hydroxy-7-methoxy groups) exhibited more effective inhibition than their analogues, scopoletin (**10**) (with 7-hydroxy-6-methoxy groups) and aesculetin dimethyl ester (**12**) (with 6,7-dimethoxy groups) against LPS-induced NO generation; (d) among the 6,7,8-trisubstituted coumarin derivatives, fraxetin (**14**) (with 7,8-dihydroxy-6-methoxy groups) displayed more effective inhibition than its analogue, fraxidin (**13**) (with 8-hydroxy-6,7-dimethoxy groups) against LPS-induced NO generation; (e) (8*E*)-4′′-*O*-Methylligstroside (**1**), aesculetin (**9**), and fraxetin (**14**) are the most effective among the isolated compounds, with IC_50_ values of 12.38 ± 0.86, 9.36 ± 0.25, and 10.11 ± 0.47 μM, respectively, against LPS-induced NO production; (e) cytotoxic effects were tested using MTT experiment. The high cell viability (95, 98, and 97 %, respectively) of compounds **1**, **9**, and **14** at 50 μM showed that their inhibitory activities against LPS-induced NO generation did not arise from their cytotoxicities.

The results of enzyme-linked immunosorbent assay (ELISA) showed that (8*E*)-4′′-*O*-methylligstroside (**1**), aesculetin (**9**), and fraxetin (**14**) obviously suppressed TNF-α and IL-6 production in a concentration-dependent manner in RAW264.7 macrophages (Figure 5). Andrographolide was used as positive control. The action mechanisms of **1**, **9**, and **14** in macrophages were further investigated. Mitogen-activated protein kinases (MAPKs) and IκBα are the downstream signaling of LPS in macrophage cell line RAW264.7. Compounds **1**, **9**, and **14** (10 μM) caused a significant reduction of the phosphorylation of MAPKs and IκBα in LPS-induced macrophages (Figure 6). Notably, phosphorylation of JNK caused by LPS was most significantly inhibited by these compounds. These results suggest that the anti-inflammatory effects of compounds **1**, **9**, and **14** are through the inhibition of activation of MAPKs and IκBα in LPS-activated macrophages.

M2-polarized macrophages are important for tissue repair [34]. Arginase 1 is an important M2 marker that connects Krüppel-like factor 4 (KLF4) to the biologic processes involved in M2 polarization [35]. High levels of arginase-1 can compete with iNOS for arginine and reduce NO production [36]. In addition, KLF4, which is one of the major members of the KLF family, was shown to induce M2 macrophage phenotype, whereas it reduced M1 macrophage expression [37]. We further examined whether compounds **1**, **9**, and **14** enhanced the expression level of M2 macrophages. The result showed that expression levels of arginase-1 and KLF4 were both induced by treatment with compounds **1**, **9**, and **14** (Figure 7). These results suggested that compounds **1**, **9**, and **14** promoted the expression of macrophage M2 markers, arginase-1 and KLF4, and exhibited the anti-inflammatory activity. We can also draw a schematic diagram that shows how compounds **1**, **9**, and **14** influence the polarization of M1 and M2 macrophages (Figure 8).

## 3. Materials and Methods

### 3.1. General Procedures

Melting points were determined on a Yanaco micro-melting point apparatus (Yanaco, Tokyo, Japan) and are uncorrected. Optical rotations were measured using a Jasco DIP-370 polarimeter (Jasco, Easton, MD, USA) in CHCl_3_. Ultraviolet (UV) spectra were obtained on a Jasco UV-240 spectrophotometer (Jasco, Easton, MD, USA). Infrared (IR) spectra (neat or KBr) were recorded on a Perkin Elmer 2000 FT-IR spectrometer (PerkinElmer, Waltham, MA, USA). Nuclear magnetic resonance (NMR) spectra, including correlation spectroscopy (COSY), nuclear Overhauser effect spectrometry (NOESY), rotating frame nuclear Overhauser effect spectrometry (ROESY), heteronuclear multiple-bond correlation (HMBC), and heteronuclear single-quantum coherence (HSQC) experiments, were acquired using a Varian Inova 500 spectrometer operating at 500 MHz (^1^H) and 125 MHz (^13^C), respectively, with chemical shifts given in ppm (δ) using tetramethylsilane (TMS) as an internal standard. Electrospray ionization (ESI) and high-resolution electrospray ionization (HRESI) mass spectra were recorded on a Bruker APEX II mass spectrometer (Bruker, Bremen, Germany). Silica gel (70–230, 230–400 mesh) (Merck, Darmstadt, Germany) was used for column chromatography (CC). Silica gel 60 F-254 (Merck, Darmstadt, Germany) was used for thin-layer chromatography (TLC) and preparative thin-layer chromatography (PTLC).

### 3.2. Plant Material

The stem bark of *F. chinensis* was collected from Pingtung County, Taiwan, in April 2011 and identified by Prof. J. J. Chen. A voucher specimen (FC 201104) was deposited in the Faculty of Pharmacy, National Yang-Ming University, Taipei, Taiwan.

### 3.3. Extraction and Isolation

The dried stem bark (4.0 kg) of *F. chinensis* was pulverized and extracted three times with MeOH (≥99%, 20 L each) for three days at room temperature. The MeOH extract was concentrated under reduced pressure at 35 °C, and the residue (384 g) was partitioned between EtOAc (≥99.5%) and H_2_O (≥99.5%) (1:1) to provide the EtOAc-soluble fraction (fraction A; 180 g). Fraction A (180 g) was purified by column chromatography (CC) (10 × 72 cm, 7.2 kg of silica gel, 70–230 mesh; CH_2_Cl_2_ (≥99%)/MeOH gradient) to afford 10 fractions: A1 (5 L, CH_2_Cl_2_), A2 (6 L, CH_2_Cl_2_/MeOH, 90:1), A3 (9 L, CH_2_Cl_2_/MeOH, 80:1), A4 (6 L, CH_2_Cl_2_/MeOH, 60:1), A5 (5 L, CH_2_Cl_2_/MeOH, 50:1), A6 (10 L, CH_2_Cl_2_/MeOH, 40:1), A7 (5 L, CH_2_Cl_2_/MeOH, 20:1), A8 (3 L, CH_2_Cl_2_/MeOH, 10:1), A9 (4 L, CH_2_Cl_2_/MeOH, 1:1), and A10 (2 L, MeOH). Fraction A1 (12.5 g) was subjected to CC (5 × 45 cm, 500 g of silica gel, 230–400 mesh; *n*-hexane/acetone (≥99%) 20:1–0:1, 500 mL fractions) to give eight subfractions: A1-1–A1-8. Part (56 mg) of fraction A1-1 was further purified by preparative TLC (silica gel; *n*-hexane (99%)/EtOAc 2:1) to afford 4-hydroxyphenethyl acetate (**22**) (6.6 mg) (R*_f_* = 0.78). Part (38 mg) of fraction A1-2 was further purified by preparative TLC (silica gel; *n*-hexane/EtOAc, 1:1) to obtain *p*-hydroxybenzaldehyde (**23**) (4.2 mg) (R*_f_* = 0.90). Part (76 mg) of fraction A1-3 was further purified by preparative TLC (silica gel; *n*-hexane/EtOAc*,* 3:1) to afford (*E*)-ferulaldehyde (**20**) (6.6 mg) (R*_f_* = 0.25). Part (105 mg) of fraction A1-5 was further purified by preparative TLC (silica gel; CH_2_Cl_2_/EtOAc, 8:1) to yield (+)-pinoresinol (**24**) (5.5 mg) (R*_f_* = 0.32) and (+)-salicifoliol (**25**) (5.6 mg) (R*_f_* = 0.26). Part (55 mg) of fraction A1-8 was purified by preparative TLC (silica gel; CH_2_Cl_2_/acetone, 30:1) to obtain α-tocopheryl quinone (**26**) (2.4 mg) (R*_f_* = 0.73). Fraction A3 (19.8 g) was subjected to CC (5 × 70 cm, 895 g of silica gel, 230–400 mesh; *n*-hexane/EtOAc 10:1–0:1, 300 mL fractions) to give 10 subfractions: A3-1–A3-10. Part (69 mg) of fraction A3-1 was further purified by preparative TLC (silica gel; *n*-hexane/acetone, 1:1) to afford isoscopoletin (**11**) (2.8 mg) (R*_f_* = 0.81). Part (125 mg) of fraction A3-3 was further purified by preparative TLC (silica gel; CH_2_Cl_2_/MeOH, 25:1) to obtain olenoside A (**4**) (7.9 mg) (R*_f_* = 0.70) and umbelliferone (**15**) (3.7 mg) (R*_f_* = 0.70). Part (71 mg) of fraction A3-4 was further purified by preparative TLC (silica gel; CH_2_Cl_2_/MeOH, 25:1) to afford fraxidin (**13**) (5.2 mg) (R*_f_* = 0.41). Part (33 mg) of fraction A3-8 was further purified by preparative TLC (silica gel; CH_2_Cl_2_/acetone, 6:1) to yield tyrosol (**21**) (4.9 mg) (R*_f_* = 0.52). Part (92 mg) of fraction A3-10 was purified by preparative TLC (silica gel; *n*-hexane/acetone, 1:2) to obtain fraxetin (**14**) (15.9 mg) (R*_f_* = 0.30). Fraction A4 (16.7 g) was subjected to CC (5 × 60 cm, 755 g of silica gel, 230–400 mesh; CH_2_Cl_2_/acetone 10:1–0:1, 1.2 L-fractions) to give eight subfractions: A4-1–A4-8. Part (290 mg) of fraction A4-2 was purified by CC (silica gel, *n*-hexane/acetone 3:2) to afford four subfractions (each 1.2 L, A4-2-1–A4-2-4). Part (43 mg) of fraction A4-2-3 was further purified by preparative TLC (silica gel; CH_2_Cl_2_/acetone 15:1) to obtain scopoletin (**10**) (3.1 mg) (R*_f_* = 0.39). Part (61 mg) of fraction A4-3 was further purified by preparative TLC (silica gel; CH_2_Cl_2_/acetone, 5:1) to afford aesculetin (**9**) (7.6 mg) (R*_f_* = 0.46). Fraction A6 (27.4 g) was subjected to CC (7 × 60 cm, 1.3 kg of silica gel, 230–400 mesh; CH_2_Cl_2_/EtOAc 10:1–0:1, 1 L-fractions) to give 11 subfractions: A6-1–A6-11. Part (61 mg) of fraction A6-1 was further purified by preparative TLC (silica gel; *n*-hexane/acetone, 5:1) to afford methyl 3,4-dimethoxycinnamate (**18**) (4.3 mg) (R*_f_* = 0.68). Part (210 mg) of fraction A6-2 was purified by CC (silica gel, *n*-hexane/acetone 5:1) to afford five subfractions (each 250 mL, A6-2-1–A6-2-5). Part (28 mg) of fraction A6-2-2 was further purified by preparative TLC (silica gel; *n*-hexane/EtOAc*,* 2:1) to afford methyl ferulate (**17**) (5.7 mg) (R*_f_* = 0.70). Part (54 mg) of fraction A6-3 was further purified by preparative TLC (silica gel; CH_2_Cl_2_/EtOAc, 6:1) to yield methyl isoferulate (**16**) (4.2 mg) (R*_f_* = 0.38). Part (56 mg) of fraction A6-4 was purified by preparative TLC (silica gel; *n*-hexane/EtOAc, 2:1) to obtain aesculetin dimethyl ester (**12**) (4.7 mg) (R*_f_* = 0.71) and methyl (*E*)-*p*-coumarate (**19**) (3.9 mg) (R*_f_* = 0.73). Part (38 mg) of fraction A6-7 was further purified by preparative TLC (silica gel; CHCl_3_ (≥99%)/MeOH*,* 8:1) to afford oleoside methyl ester (**8**) (5.7 mg) (R*_f_* = 0.19). Fraction A9 (17.3 g) was subjected to CC (5 × 60 cm, 780 g of silica gel, 230–400 mesh; CH_2_Cl_2_/MeOH 7:1–0:1, 500 mL fractions) to give 13 subfractions: A9-1–A9-13. Part (75 mg) of fraction A9-4 was further purified by preparative TLC (silica gel; CHCl_3_/MeOH, 7:1) to afford (8*E*)-4′′-*O*-methylligstroside (**1**) (15.7 mg) (R*_f_* = 0.66). Part (63 mg) of fraction A9-5 was further purified by preparative TLC (silica gel; CHCl_3_/MeOH, 5:1) to yield (8*E*)-3′′,4′′-di-*O*-methyloleuropein (**7**) (7.4 mg) (R*_f_* = 0.53). Part (108 mg) of fraction A9-7 was purified by CC (silica gel, CHCl_3_/MeOH, 3:1) to afford three subfractions (each 150 mL, A9-7-1–A9-7-3). Fraction A9-7-1 (63 mg) was further purified by preparative TLC (silica gel; CHCl_3_/MeOH, 4:1) to obtain 3′′,4′′-di-*O*-methyldemethyloleuropein (**3**) (6.1 mg) (R*_f_* = 0.61). Fraction A9-7-2 (28 mg) was further purified by preparative TLC (silica gel; CHCl_3_/MeOH, 4:1) to afford (8*E*)-4′′-*O*-methyldemethylligstroside (**2**) (7.2 mg) (R*_f_* = 0.55). Fraction A10 (36.8 g) was subjected to silica gel column chromatography (10 × 55 cm, 230–400 mesh) with CH_2_Cl_2_/MeOH, 7:1 to give 14 fractions (each 1.5 L). Part (115 mg) of fraction 10-4 was purified further by preparative TLC (silica gel, CHCl_3_/MeOH, 7:1) to afford (8*E*)-ligstroside (**5**) (12.3 mg) (R*_f_* = 0.65). Part (132 mg) of fraction 10-5 was purified further by preparative TLC (silica gel, CH_2_Cl_2_/acetone, 1:2) to yield oleuropein (**6**) (14.6 mg) (R*_f_* = 0.35).

*(8E)-4′′-O-Methylligstroside* (**1**): yellowish oil; [α]^25^_D_: −182.2 (*c* 0.2, MOH); UV (MeOH): λ_max_ (log ε) = 238 (4.05), 276 (3.82), 283 (3.81), 318 (3.76) nm; IR (neat): υ_max_ = 3402 (OH), 1727 (C=O), 1708 (C=O) cm^−1^; ESI-MS: *m/z* = 561 [M + Na]^+^; HR-ESI-MS: *m/z* = 561.1950 [M + Na]^+^ (calcd. for C_26_H_34_O_12_Na: 561.1948). ^1^H and ^13^C NMR spectroscopic data, see Table 1.

*(8E)-4′′-O-Methyldemethylligstroside* (**2**): yellowish oil; [α]^25^_D_: −181.5 (*c* 0.25, MOH); UV (MeOH): λ_max_ (log ε) = 225 (4.04), 276 (3.80), 282 (3.79), 317 (3.73) nm; IR (neat): υ_max_ = 3334 (OH), 1728 (C=O), 1707 (C=O) cm^−1^; ESI-MS: *m/z* = 547 [M + Na]^+^; HR-ESI-MS: *m/z* = 547.1787 [M + Na]^+^ (calcd. for C_25_H_32_O_12_Na: 547.1791). ^1^H and ^13^C NMR spectroscopic data, see Table 1.

*3′′,4′′-Di-O-methyldemethyloleuropein* (**3**): amorphous powder; [α]^25^_D_: −155.2 (*c* 0.22, MOH); UV (MeOH): λ_max_ (log ε) = 226 (4.24), 277 (3.40) nm; IR (neat): υ_max_ = 3350 (OH), 1721 (C=O), 1698 (C=O) cm^−1^; ESI-MS: *m/z* = 577 [M + Na]^+^; HR-ESI-MS: *m/z* = 577.1892 [M + Na]^+^ (calcd. for C_26_H_34_O_13_Na: 577.1897). ^1^H and ^13^C NMR spectroscopic data, see Table 1.

### 3.4. Biological Assay

The activity of the isolated compounds on neutrophil and macrophage proinflammatory response was evaluated by monitoring the inhibition of all isolated compounds against fMLP/CB-induced O_2_^•−^ and elastase release and against LPS-induced NO generation in a concentration-dependent manner.

#### 3.4.1. Mensuration of Human Neutrophils

Human neutrophils from venous blood of adult, healthy volunteers (20–27 years old) were isolated by a standard pattern of dextran sedimentation before centrifugation in a Ficoll Hypaque gradient and hypotonic lysis of erythrocytes [38]. Purified neutrophils having >98% viable cells, as detected by the trypan blue exclusion method [39], were resuspended in a calcium (Ca^2+^)-free HBSS buffer at pH 7.4 and were kept at 4 °C prior to use.

#### 3.4.2. Mensuration of Superoxide Anion (O_2_^•−^) Generation

The assay for measurement of O_2_^•−^ generation was based on the SOD-inhibitable reduction of ferricytochrome *c* [40,41]. In short, after supplementation with 1 mM Ca^2+^ and 0.5 mg/mL ferricytochrome *c*, neutrophils (6 × 10^5^/mL) were equilibrated at 37 °C for 2 min and incubated with varied concentrations (10–0.01 μg/mL) of DMSO (as control) or tested compounds **1**–**26** for 5 min. Cells were incubated with cytochalasin B (1 μg/mL) for 3 min before the activation with 100 nM formyl-L-methionyl-L-leucyl-L-phenylalanine for 10 min. Changes in absorbance with the reduction of ferricytochrome *c* at 550 nm were constantly detected in a double-beam, six-cell positioner spectrophotometer with continuous stirring (Hitachi U-3010, Tokyo, Japan). Calculations were founded on differences in the reactions with and without SOD (100 U/mL) divided by the extinction coefficient for the reduction of ferricytochrome *c* (ε = 21.1/mM/10 mm).

#### 3.4.3. Measurement of Elastase Release

Degranulation of azurophilic granules was measured by determining elastase release as reported previously [41,42]. Assays were carried out applying MeO-Suc-Ala-Ala-Pro-Val-*p*-nitroanilide as elastase substrate. In brief, after supplementation with MeO-Suc-Ala-Ala-Pro-Val-*p*-nitroanilide (100 μM), neutrophils (6 × 10^5^/mL) were equilibrated at 37 °C for 2 min and incubated with tested compounds for 5 min. Cells were treated with fMLP (100 nM)/CB (0.5 μg/mL), and the changes in absorbance at 405 nm were detected constantly in order to measure elastase release. The results were displayed as the percent of elastase release in the fMLP/CB-activated, drug-free control system.

#### 3.4.4. Determination of NO Production

NO production was indirectly assessed by measuring the nitrite levels in the cultured media and serum determined by a colorimetric method based on the Griess reaction. RAW264.7 cells were pretreated with compounds for 1 h, and then stimulated with LPS (100 ng/mL) for 20 h at 37 °C. Then, cells were dispensed into 96-well plates, and 100 μL of each supernatant was mixed with the same volume of Griess reagent (1% sulfanilamide, 0.1% naphthylethylenediamine dihydrochloride, and 5% phosphoric acid) and incubated at room temperature for 10 min; the absorbance was measured at 540 nm with a Micro-Reader (Molecular Devices). By using sodium nitrite to generate a standard curve, the concentration of nitrite was measured from absorbance at 540 nm [43].

#### 3.4.5. Cell Viability Assay

Cells (4 × 10^5^) were cultured in 96-well plates containing DMEM supplemented with 10% FBS for one day to become nearly confluent. Then, cells were cultured with compounds **1**–**26** in the presence of 100 ng/mL LPS (lipopolysaccharide) for 24 h. After that, the cells were washed twice with DPBS and incubated with 100 μL of 0.5 mg/mL MTT for 2 h at 37 °C testing for cell viability. The medium was then discarded and 100 μL dimethyl sulfoxide (DMSO) was added. After 30 min incubation, absorbance at 570 nm was read using a microplate reader (Molecular Devices, Sunnyvale, CA, USA) [44].

#### 3.4.6. Enzyme-Linked Immunosorbent Assay

RAW264.7 cells (4 × 10^5^ cells in 96-well plates) were pretreated with compounds **1**, **9**, **14**, or vehicle (0.05% DMSO) for 1 h and then stimulated with LPS (100 ng/mL) for 20 h. Supernatants were collected and analyzed for production of TNF-α and IL-6 by using appropriate ELISA kits (R&D, MN, USA) in accordance with the manufacturer’s instructions.

#### 3.4.7. Western Blot

Western blot analysis followed as previously described with slight changes [45]. Cells (1.0 × 10^6^) were seeded into 6 cm dishes and grown until 80–85% confluent. RAW264.7 cells were pretreated with **1**, **9**, and **14** (10 μM) for 6 h, and then stimulated with LPS (100 ng/mL) for 15 min (for detecting p-IκBα, p-ERK, p-JNK, and p-p38) or 20 h (for detecting arginase 1 and KLF4) at 37 °C. Cultured medium was removed and cells were washed with ice-cold PBS. After RIPA buffer (Cell Signaling, MA, USA) was added, cells were scraped off the plate and transferred to the Eppendorf on ice immediately. The proteins were quantified using the BCA protein assay. Cells were preserved at −80 °C overnight and then centrifuged (15,000 × rpm, 30 min, 4 °C). Equal amounts of protein samples (25 μg) and prestained protein marker were loaded onto SDS-PAGE. After being stacked at 80 V and separated at 100 V, the proteins were transferred onto the polyvinylidene fluoride (PVDF) membranes at 350 mA. The PVDF membranes were blocked with 5% (*w/v*) of BSA with Tris-buffered saline (TBST) containing 0.1% (*v/v*) Tween 20 at room temperature for 1 h and washed three times with TBST for 15 min each time. Primary antibodies were incubated with the membranes overnight, shaking at 4 °C. Then, each membrane was washed with TBST and incubated with horseradish peroxidase (HRP)-conjugated secondary antibodies at room temperature for 1 h while shaking. Finally, each membrane was developed using an ECL detection kit, and the images were visualized by ImageQuant LAS 4000mini (GE Healthcare, MA, USA). Images were quantified using Image J version 1.48 (NIH, Bethesda, MD, USA).

#### 3.4.8. Statistical Analysis

Results are expressed as the mean ± SEM, and comparisons were made using Tukey’s HSD test. A probability of 0.05 or less was considered significant. The software SigmaPlot was used for the statistical analysis.

## 4. Conclusions

Twenty-six compounds, including three undescribed secoiridoid glucosides, (8*E*)-4′′-*O*-methylligstroside (**1**), (8*E*)-4′′-*O*-methyldemethylligstroside (**2**), and 3′′,4′′-di-*O*-methyldemethyl-oleuropein (**3**), were isolated from stem bark of *F. chinensis*. The structures of these isolates were elucidated according to spectroscopic data. The effects on neutrophil proinflammatory responses of isolates were evaluated by suppressing fMLP/CB-induced O_2_^•−^ generation and elastase release by human neutrophils. The results of anti-inflammatory assays show that compounds **1**, **9**, **11**, **14**, **21**, and **22** can obviously inhibit fMLP-induced O_2_^•−^ generation and/or elastase release. (8*E*)-4′′-*O*-Methylligstroside (**1**) and fraxetin (**14**) were the most effective among the isolated compounds, with IC_50_ values of 0.08 ± 0.01 and 0.50 ± 0.10 μg/mL, respectively, against fMLP-induced O_2_^•−^ generation and elastase release. Furthermore, compounds **9** and **14** showed potent inhibition with IC_50_ values of 9.36 ± 0.25 and 10.11 ± 0.47 μM, respectively, against LPS-induced NO generation. Compounds **1**, **9**, and **14** suppressed LPS-induced NO, TNF-α, and IL-6 generation via blocking the phosphorylation of MAPKs and degradation of IκBα. In addition, compounds **1**, **9**, and **14** stimulated anti-inflammatory M2 phenotype by elevating the expression of arginase 1 and KLF4. In conclusion, compounds **1**, **9**, and **14** interfered with multiple intracellular targets. Our research indicates *F. chinensis* and its constituents (especially **1**, **9**, and **14**) may deserve further investigation as potential candidates for the treatment or prevention of various inflammatory diseases.

## Figures and Tables

**Figure 1 molecules-25-05911-f001:**
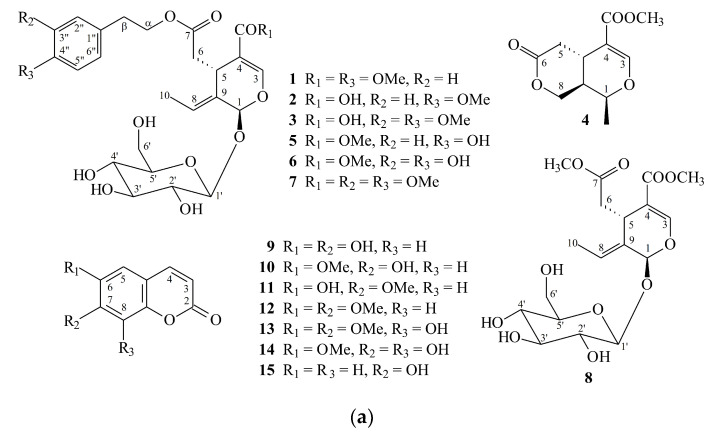
The chemical structures of compounds **1**–**15** (**a**) and **16**–**26** (**b**) isolated from *F. chinensis.*

**Figure 2 molecules-25-05911-f002:**
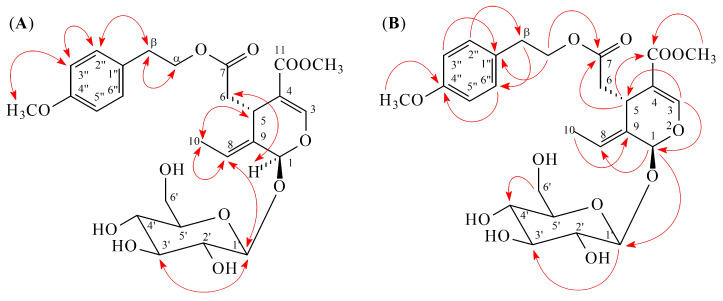
Key NOESY (**A**) and HMBC (**B**) correlations of **1**.

**Figure 3 molecules-25-05911-f003:**
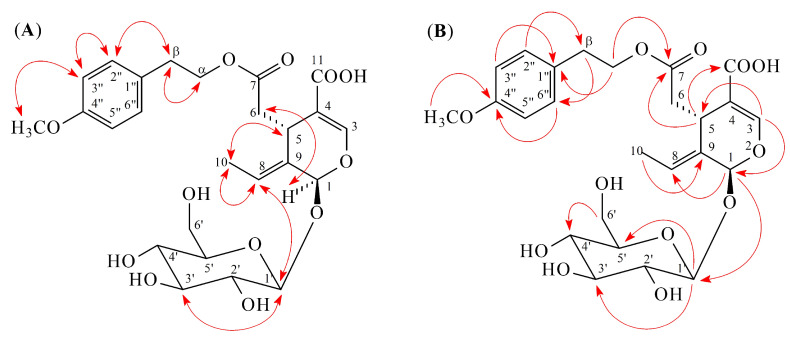
Key NOESY (**A**) and HMBC (**B**) correlations of **2**.

**Figure 4 molecules-25-05911-f004:**
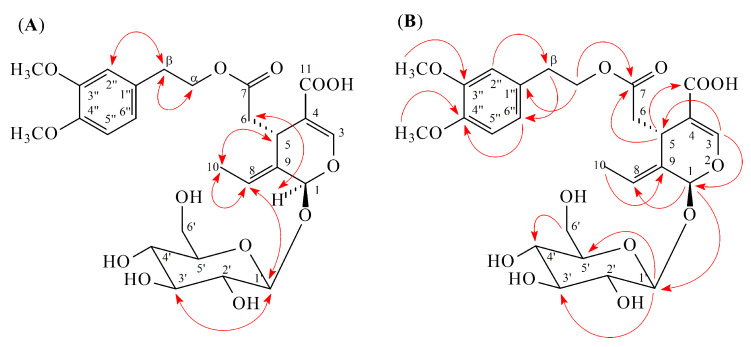
Key NOESY (**A**) and HMBC (**B**) correlations of **3**.

**Figure 5 molecules-25-05911-f005:**
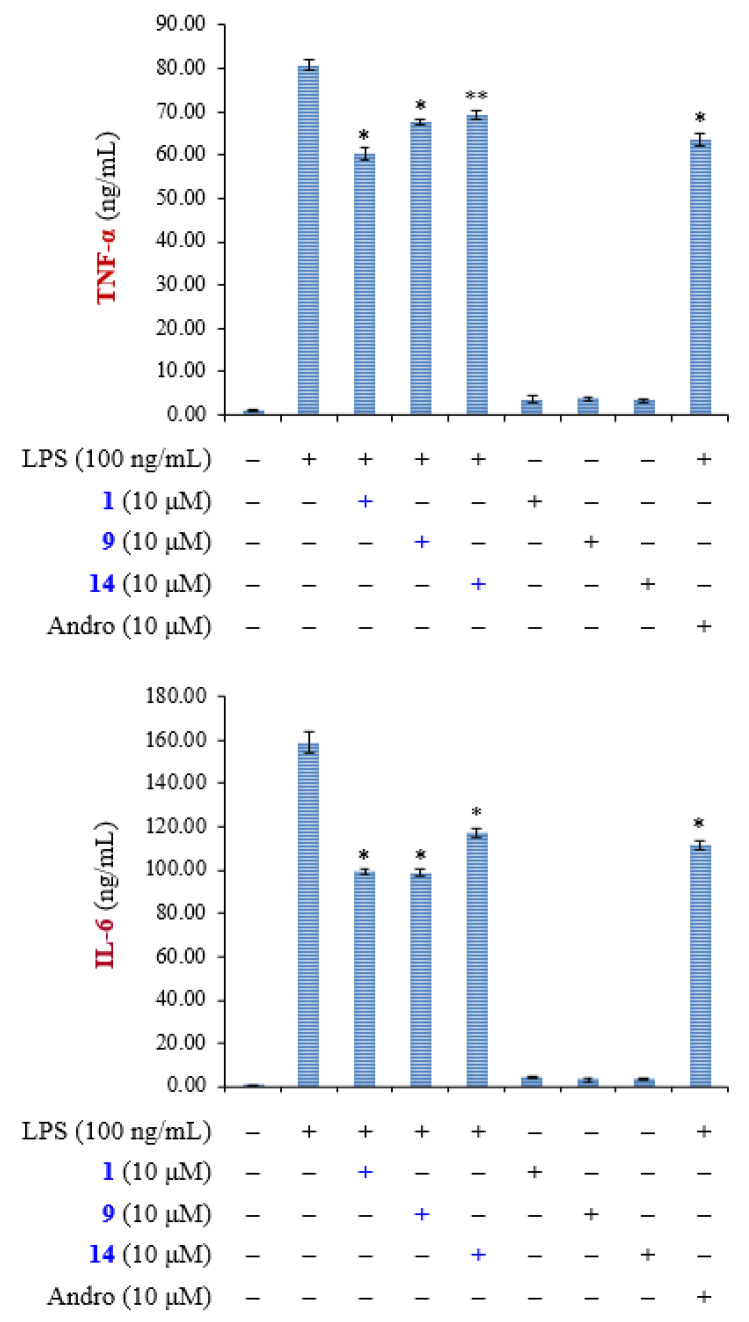
Compounds **1**, **9**, and **14** suppress the production of proinflammatory cytokines TNF-α and IL-6 in LPS-stimulated RAW 264.7 macrophages. Andrographolide (Andro) was used as positive control. Results are displayed as mean ± SEM (*n* = 3) of three independent experiments. “+” means treatment with LPS or compound. “−” means no treatment with LPS or compound. Asterisks indicate significant differences (* *p* < 0.05, ** *p* < 0.01) compared with the control.

**Figure 6 molecules-25-05911-f006:**
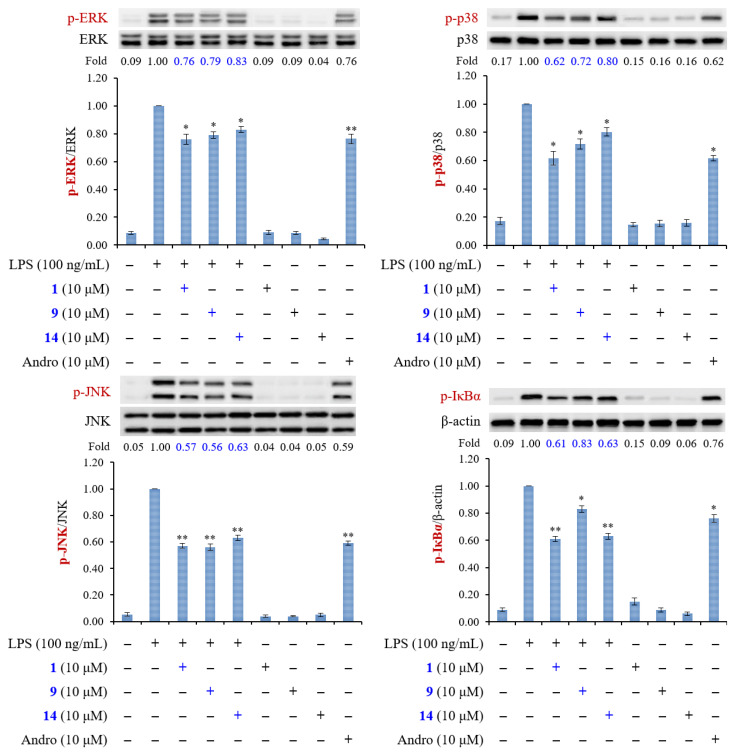
Compounds **1**, **9**, and **14** inhibit the phosphorylation of MAPKs and IκBα in LPS-activated macrophages. RAW264.7 cells were pretreated with **1**, **9**, and **14** (10 μM) for 6 h, and then stimulated with LPS for 15 min. Phosphorylation of MAPKs and IκBα was analyzed by immunoblotting. Densitometric analysis of all samples was normalized to the corresponding total protein or β-actin. Andrographolide (Andro) was used as positive control. Results are displayed as mean ± SEM of three independent experiments. “+” means treatment with LPS or compound. “−” means no treatment with LPS or compound. Asterisks indicate significant differences (* *p* < 0.05 and ** *p* < 0.01) compared with the control.

**Figure 7 molecules-25-05911-f007:**
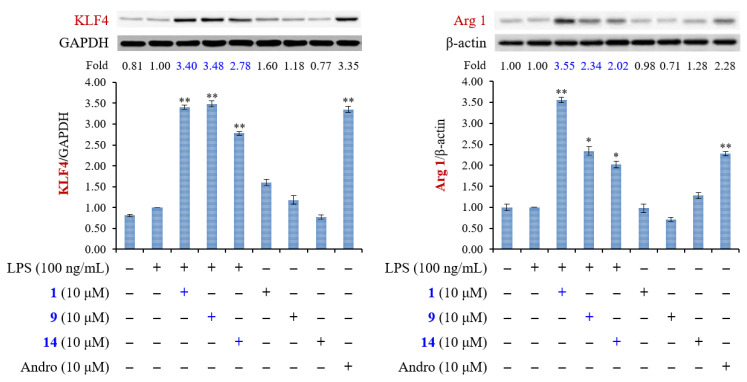
The effect of compounds **1**, **9**, and **14** on M2 polarized macrophages in LPS-stimulated RAW264.7 macrophages. RAW264.7 cells were pretreated with **1**, **9**, and **14** (10 μM) for 6 h, and then stimulated with LPS for 20 h. Expression of KLF4 and arginase 1 (Arg 1) were determined by Western blot analysis. Andrographolide (Andro) was used as positive control. The data were expressed as mean ± SEM of three independent experiments. “+” means treatment with LPS or compound. “−” means no treatment with LPS or compound. Asterisks indicate significant differences (* *p* < 0.05 and ** *p* < 0.01) compared with the control.

**Figure 8 molecules-25-05911-f008:**
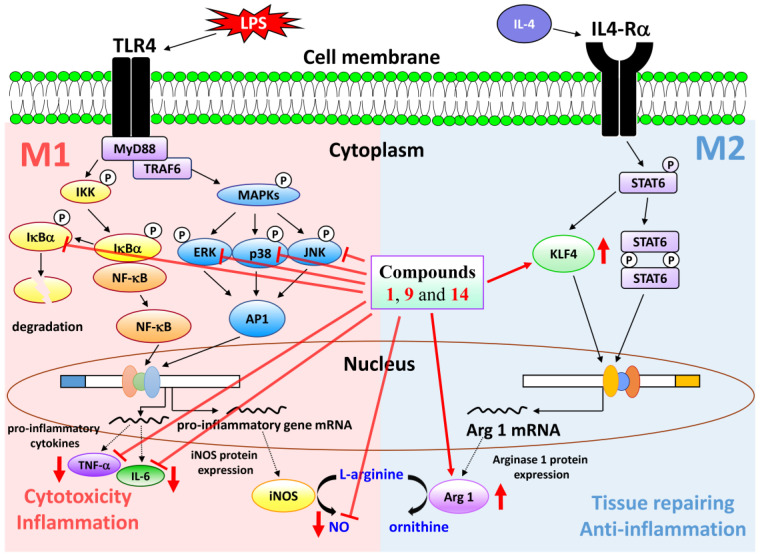
Schematic diagram for anti-inflammatory action of compounds **1**, **9**, and **14** in LPS-induced RAW264.7 macrophages.

**Table 1 molecules-25-05911-t001:** ^1^ H-NMR data for compounds **1**–**3** (δ in ppm, *J* in Hz).

Position	1 ^a^	2 ^a^	3 ^a^
1	5.92 br s	5.87 br s	5.86 br s
3	7.51 s	7.39 s	7.38 s
5	3.95 dd (9.5, 5.0)	4.01 dd (9.5, 5.0)	4.00 dd (9.5, 4.5)
6	2.44 dd (14.0, 9.5)	2.41 dd (14.0, 9.5)	2.41 dd (14.0, 9.5)
	2.69 dd (14.0, 5.0)	2.79 dd (14.0, 4.5)	2.80 dd (14.0, 4.5)
8	6.06 br q (7.0)	6.05 br q (7.0)	6.06 br q (7.0)
10	1.62 dd (7.0, 1.0)	1.66 dd (7.0, 1.0)	1.66 dd (7.0, 1.0)
α	4.12 dt (10.5, 7.0)	4.11 dt (10.5, 7.0)	4.15 dt (10.5, 7.0)
	4.24 dt (10.5, 7.0)	4.23 dt (10.5, 7.0)	4.27 dt (10.5, 7.0)
β	2.85 t (7.0)	2.85 t (7.0)	2.86 t (7.0)
1′	4.80 d (7.5)	4.80 d (8.0)	4.80 d (8.0)
2′	3.28–3.36 m	3.29–3.35 m	3.28–3.36 m
3′	3.42 dd (8.5, 8.5)	3.41 dd (9.0, 8.5)	3.41 dd (8.5, 8.5)
4′	3.28–3.36 m	3.29–3.35 m	3.28–3.36 m
5′	3.28–3.36 m	3.29–3.35 m	3.28–3.36 m
6′	3.66 dd (12.0, 5.5)	3.67 dd (12.0, 5.5)	3.66 dd (12.0, 5.5)
	3.88 dd (12.0, 1.5)	3.88 dd (12.0, 1.5)	3.88 dd (12.0, 1.5)
2′′	7.15 d (9.0)	7.15 d (9.0)	6.86 d (2.0)
3′′	6.85 d (9.0)	6.85 d (9.0)	
5′′	6.85 d (9.0)	6.85 d (9.0)	6.88 d (8.5)
6′′	7.15 d (9.0)	7.15 d (9.0)	6.79 dd (8.5, 2.0)
OMe-11	3.71 s		
OMe-3′′			3.82 s
OMe-4′′	3.76 s	3.76 s	3.80 s

^a^ Measured in CD_3_OD at 500 MHz.

**Table 2 molecules-25-05911-t002:** ^13^C-NMR data for compounds **1**–**3** (δ in ppm).

Position	1 ^a^	2 ^a^	3 ^a^
1	95.3	95.0	95.3
3	155.3	152.8	155.3
4	109.5	110.2	109.5
5	32.0	31.9	32.0
6	41.3	41.3	41.4
7	173.4	173.3	173.4
8	125.1	124.6	125.0
9	130.5	130.7	130.6
10	13.7	13.6	13.7
11	168.9	171.0	168.8
α	67.0	66.9	66.9
β	35.2	35.2	35.7
1′	101.0	101.0	101.0
2′	74.9	74.9	74.9
3′	78.1	78.1	78.1
4′	71.7	71.6	71.6
5′	78.6	78.6	78.6
6′	62.9	62.9	62.9
1′′	131.5	131.5	132.4
2′′	131.2	131.2	114.1
3′′	115.1	115.1	150.5
4′′	160.0	160.0	149.3
5′′	115.1	115.1	113.4
6′′	131.2	131.2	122.6
OMe-11	52.1		
OMe-3′′			56.7
OMe-4′′	55.9	55.9	56.7

^a^ Measured in CD_3_OD at 125 MHz.

**Table 3 molecules-25-05911-t003:** Inhibitory effects of compounds **1**–**26** from the stem bark of *F*. *chinensis* on superoxide radical anion generation and elastase release by human neutrophils in response to fMet-Leu-Phe/cytochalasin B ^a^.

Compounds	Superoxide Anion	Elastase
	IC_50_ [µg/mL] ^b^ or (Inh %) ^c^	
(8*E*)-4′′-*O*-Methylligstroside (**1**)	0.08 ± 0.01 ***	2.57 ± 0.76 ***
(8*E*)-4′′-*O*-Methyldemethylligstroside (**2**)	2.66 ± 0.33 ***	(42.92 ± 4.45) ***
3′′,4′′-Di-*O*-methyldemethyloleuropein (**3**)	5.22 ± 2.34 ***	(33.78 ± 1.64) ***
Olenoside A (**4**)	(8.67 ± 1.62) **	(19.87 ± 2.94) **
(8*E*)-Ligstroside (**5**)	(1.30 ± 1.88)	(26.58 ± 3.94) **
Oleuropein (**6**)	2.90 ± 0.46	(23.76 ± 0.50) ***
(8*E*)-3′′,4′′-Di-*O*-methyloleuropein (**7**)	(11.34 ± 6.05) *	(32.73 ± 4.35) **
Jaspolyside methyl ester (**8**)	(14.39 ± 3.28) *	(20.54 ± 2.24) ***
Esculetin (**9**)	0.17 ± 0.03	2.41 ± 0.60
Copoletin (**10**)	(−0.91 ± 1.16)	(8.98 ± 1.68) **
Soscopoletin (**11**)	5.20 ± 1.52	3.23 ± 0.68
Esculetin dimethyl ester (**12**)	7.65 ± 1.62	(8.95 ± 2.94) *
Raxidin (**13**)	(8.95 ± 2.94) *	(12.14 ± 1.91) **
Raxetin (**14**)	0.19 ± 0.01	0.50 ± 0.10
Umbelliferone (**15**)	(3.38 ± 1.99)	(27.92 ± 4.88)
Methyl isoferulate (**16**)	(9.03 ± 1.65) **	(-2.76 ± 0.84) *
Methyl ferulate (**17**)	(23.02 ± 4.18) **	(24.12 ± 4.58) **
Methyl 3,4-dimethoxycinnamate (**18**)	(42.90 ± 3.97) ***	(7.05 ± 0.68) ***
Methyl (*E*)-*p*-coumarate (**19**)	(8.01 ± 0.66) ***	(20.30 ± 3.37) **
(*E*)-Ferulaldehyde (**20**)	(31.40 ± 7.95) **	(38.61 ± 3.64) ***
Tyrosol (**21**)	4.93 ± 0.19	2.64 ± 0.22
4-Hydroxyphenethyl acetate (**22**)	2.50 ± 0.35	3.03 ± 0.48
*p*-Hydroxybenzaldehyde (**23**)	(16.16 ± 2.03) **	(24.35 ± 4.45) **
(+)-Pinoresinol (**24**)	2.01 ± 0.38	(42.37 ± 2.06) ***
(+)-Salicifoliol (**25**)	(3.70 ± 2.59)	(9.14 ± 1.58) **
α-Tocopheryl quinone (**26**)	(17.88 ± 2.82) **	(11.35 ± 4.41)
Diphenyleneiodonium (DPI) ^d^	0.52 ± 0.19 ***	-
Phenylmethylsulfonyl fluoride (PMSF) ^d^	-	34.4 ± 5.1 ***

^a^ Results are displayed as mean ± SEM (*n* = 3) of three independent experiments. ^b^ Concentration necessary for 50% inhibition (IC_50_). If IC_50_ value of tested compound was <10 μg/mL, it was presented as IC_50_ [μg/mL]. ^c^ Percentage of inhibition (Inh %) at 10 μg/mL. If IC_50_ value of tested compound was ≥10 μg/mL, it was displayed as Inh % at 10 μg/mL. ^d^ DPI and PMSF were employed as positive controls for superoxide anion (O_2_^•−^) production and elastase release, respectively. * *p* < 0.05 compared with the control. ** *p* < 0.01 compared with the control. *** *p* < 0.001 compared with the control.

**Table 4 molecules-25-05911-t004:** Inhibitory effects of compounds **1**–**26** from the stem bark of *F*. *chinensis* on nitric oxide (NO) generation by RAW264.7 murine macrophages in response to lipopolysaccharide (LPS).

Compounds	IC_50_ [μM] ^a^
(8*E*)-4′′-*O*-Methylligstroside (**1**)	12.38 ± 0.86 *
(8*E*)-4′′-*O*-Methyldemethylligstroside (**2**)	24.72 ± 1.25 **
3′′,4′′-Di-*O*-methyldemethyloleuropein (**3**)	37.14 ± 2.51 *
Olenoside A (**4**)	>100
(8*E*)-Ligstroside (**5**)	42.78 ± 3.23 *
Oleuropein (**6**)	40.02 ± 2.69 *
(8*E*)-3′′,4′′-Di-*O*-methyloleuropein (**7**)	53.44 ± 4.19
Jaspolyside methyl ester (**8**)	65.82 ± 5.64
Esculetin (**9**)	9.36 ± 0.25 **
Copoletin (**10**)	53.05 ± 3.63 *
Soscopoletin (**11**)	15.36 ± 0.81 *
Esculetin dimethyl ester (**12**)	31.80 ± 2.17 *
Raxidin (**13**)	50.62 ± 3.08 *
Raxetin (**14**)	10.11 ± 0.47 *
Umbelliferone (**15**)	48.24 ± 3.22
Methyl isoferulate (**16**)	>100
Methyl ferulate (**17**)	>100
Methyl 3,4-dimethoxycinnamate (**18**)	75.84 ± 6.28
Methyl (*E*)-*p*-coumarate (**19**)	>100
(*E*)-Ferulaldehyde (**20**)	67.38 ± 4.09
Tyrosol (**21**)	27.11 ± 1.87 *
4-Hydroxyphenethyl acetate (**22**)	35.36 ± 2.54 *
*p*-Hydroxybenzaldehyde (**23**)	55.13 ± 4.25
(+)-Pinoresinol (**24**)	41.69 ± 3.02 *
(+)-Salicifoliol (**25**)	>100
α-Tocopheryl quinone (**26**)	>100
Quercetin ^b^	33.95 ± 2.34 *

^a^ The IC_50_ values were calculated from the slope of the dose–response curves (SigmaPlot). Values are expressed as mean ± SEM (*n* = 4) of three independent experiments. * *p* < 0.05, ** *p* < 0.01 compared with the control. ^b^ Quercetin was used as a positive control.

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
