# Peer review of "Secoiridoid Glucosides and Anti-Inflammatory Constituents from the Stem Bark of Fraxinus chinensis"

_molecules, 2020, doi:10.3390/molecules25245911_

Round 1

Reviewer 1 Report

The first time a species is mentioned (in the Abstract; it is in the Introduction section-ok), the person responsible for the first classification must be named.

Materials:

Please, specify, in the NMR section, the experiments used and the important parameters. The same is true for MS section. Both should be better specified.

Please, state the amount of plants collected and the sampling strategy. Why did the authors did not resource to use a more efficient technique for the first extraction? 60 Liters in 3 days can lead to lots of artifacts. Since there is no information that support that this is a representative sample for this species and after these 3 days in contact with Methanol, there is no unequivocal proof that these compounds are in fact constitutive of this taxon. Of course, this is me “playing the devil’s advocate” but there is truth in this, and it be part of the next studies. The Journals should emphasize this primary steps.

Please, state the purity of the Methanol and other solvents used.

Did you really used 7.2 Kg of silica gel? Maybe more important than the amount of silica gel, to guarantee reproducibility, authors should always state the dimensions of the columns and the volume used for solvents. The fractions, thus, will be reported as elution volume.

Note that, by following this approach, the authors ended up spending too much effort and material; not to mention about the waste produced is huge. Thus, I charge the authors to state why they decided to follow such an expensive method to identify many known compounds.

The NMR data should be submitted into an open-access database for a world-wide F.A.I.R. advance of science. Personally, I suggest nmrshiftdb (nmrshiftdb.nmr.uni-koeln.de/); consider this https://pubs.rsc.org/en/content/articlelanding/2019/fd/c8fd00227d#!divAbstract. 

Regarding the structural elucidation section, the spectra which are all as SI must be annotated for them to be useful in understanding the text in the manuscript. Please, provide the numbered structure (and Molecules as a Journal should always require that as mandatory) for every compound, the key correlations in the 2D spectra should be annotated and tables showing the comparison of the NMR data with the authentic samples from the literature data for the known compounds. Those should also be submitted to open-access database.

I will not provide any comments on the biological aspect of this manuscript since I do not feel comfortable to do it.

Author Response

Please see an attached file.

Reviewer 2 Report

Review Manuscript ID: molecules-1019577

Title: Secoiridoid Glucosides and Anti-inflammatory Constituents from the Stem Bark of Fraxinus chinensis.

  • Comments: Some acronyms should be explained (e.g. COSY, HSQC, HMBC, NOESY, HR-ESI-MS)
  • Comments: Page 3, Line 83: 32.0 (C-5) instead of 32.0 (C-),5
  • Comments: structure of 1 was elucidated as (8E)-4′′-O-methylligstroside: Explain the (E) configuration at C-8 using NOESY correlation between H3-10 and H-5 (Same comments go also with compound 2 and 3)

Author Response

See an attached file.

Reviewer 3 Report

The authors of the manuscript “Secoiridoid Glucosides and Anti-inflammatory Constituents from the Stem Bark of Fraxinus chinensis” aim to elucidate the mechanisms how the herbal supplement Qin Pin, the bark of Fraxinus chinensis, impacts on inflammation.

To study the mechanisms, the authors firstly isolate 26 compounds from the bark, including 3 new compounds, and provide their chemical structures. Furthermore, they test the capability of those compounds to reduce superoxide anion as well as nitric oxide generation. After that, 3 compounds were tested in more detail (compound 1, 9, and 14) and their impact on LPS induced TNFα and IL-6 production. Lastly, the effect of these 3 substances on the relevant signaling pathways.

The study is well designed, however, the manuscript leaves some open questions.

Major points:

  • Please explain why you picked the compounds 1, 9, and 14 for the detailed experiments. Is it because they were shown to be not cytotoxic (line 201)? What were the cytotoxic effects of the other compounds? Were they all tested?
  • How was the survival in case of the neutrophils? Please provide the data of all MTT tests.
  • The authors chose andrographolide as a positive control in Figures 5 and 6, but it is not mentioned in the manuscript/method description.
  • Why did the authors choose DPI, PMSF, qercetin and andrographolide as a positive control over some more common immune modulators?
  • Figure 7: This result is a key piece of the story. However, the figure shows the result and quantification of a single western blot. Please repeat the experiment at least twice independently with new lysates to be able to draw any conclusion from it.
  • The authors state that the compounds exhibit anti-inflammatory effects in the macrophages. Since macrophages can also be polarized into anti-inflammatory M2-like macrophages, it would be interesting to analyse if anti-inflammatory markers are induced or if generally all signaling pathways/gene expression are blocked by the compound. Interesting M2-markers would be Arginase 1 or Ym1.
  • Methods: For the results of superoxide supression, eleastase secretion as well as NO generation please provide information on the experimental setup. How long were the cells treated? How long were they kept in culture in total? Were the substances present throughout the experiment or washed out at some point?
  • Methods: For the LPS stimulation the authors state that the macrophages were pre-treated with the compounds for 1 h before being stimulated with LPS. Were the substances washed out after the pre-incubation?

Minor points:

  • In the methods section, the biological assay methods are not well described and not all citations are correct. For example; 3.4.2 (Mensuration of Human neutrophils) cites a paper working with rat hepatocytes and in 3.4.5 it is written that the method was modified, but not how. Please give a brief description of all methods that were performed.
  • Table 3 and 4 and figure 5-7: please indicate the number of samples and the number of independent experiments

Author Response

Please see an attached file.

Round 2

Reviewer 3 Report

The revised manuscript „Secoiridoid Glucosides and Anti-inflammatory Constituents from the Stem Bark of Fraxinus chinensis” by Hao-Chiun Chang et al. was greatly improved by the authors.

The new experiments showing a shift in the polarization to M2 macrophages are convincing and underline the regulatory capacity of the tested substances. The revised descriptions of the experiments are now mostly clear and understandable.

Minor comments:

  • Figure legend of Figure 6: The sentence “Andrographolide was used as positive control” was pasted twice
  • Figure legend of Figure 7: The legend states that the cells were pre-incubated for 6 hours with the compounds and then stimulated with LPS for 20 min. In the methods section it is stated that compounds and LPS were added together and the LPS-stimulation was 15 min. What is correct?
  • Figure legend of Figure 8: Please also briefly describe the scheme of stimulation as in figure legend 7 (how long was the pre-treatment and the stimulation with LPS)
  • Figure legend of Figure 8 ends with two dots (..)
  • In the methods section: Pre-treatment with the substances was only described for 3.4.2, 3.4.3 (5 min each), and 3.4.6 (1 hour). Were there no pre-incubations in the other methods? Please indicate clearly for each method if there was a pre-incubation

Author Response

Please see an attached file.
